# Peculiarities of the Spatial and Electronic Structure of 2-Aryl-1,2,3-Triazol-5-Carboxylic Acids and Their Salts on the Basis of Spectral Studies and DFT Calculations

**DOI:** 10.3390/ijms241814001

**Published:** 2023-09-12

**Authors:** Mauricio Alcolea Palafox, Nataliya P. Belskaya, Irena P. Kostova

**Affiliations:** 1Departamento de Química Física, Facultad de Ciencias Químicas, Universidad Complutense, 28040 Madrid, Spain; 2Department of Technology for Organic Synthesis, Ural Federal University, 19 Mira Str., 620012 Yekaterinburg, Russia; n.p.belskaya@urfu.ru; 3Department of Chemistry, Faculty of Pharmacy, Medical University, 2 Dunav Str., 1000 Sofia, Bulgaria; irenakostova@yahoo.com

**Keywords:** 1,2,3-triazoles, anticancer drugs design, vibrational analysis, scaling, dimer calculation

## Abstract

The molecular structure and vibrational spectra of six 1,2,3-triazoles-containing molecules with possible anticancer activity were investigated. For two of them, the optimized geometry was determined in the monomer, cyclic dimer and stacking forms using the B3LYP, M06-2X and MP2 methods implemented in the GAUSSIAN-16 program package. The effect of the *para*-substitution on the aryl ring was evaluated based on changes in the molecular structure and atomic charge distribution of the triazole ring. An increment in the positive N_4_ charge was linearly related to a decrease in both the aryl ring and the carboxylic group rotation, with respect to the triazole ring, and by contrast, to an increment in the pyrrolidine ring rotation. Anionic formation had a larger effect on the triazole ring structure than the electronic nature of the different substituents on the aryl ring. Several relationships were obtained that could facilitate the selection of substituents on the triazole ring for their further synthesis. The observed IR and Raman bands in the solid state of two of these compounds were accurately assigned according to monomer and dimer form calculations, together with the polynomic scaling equation procedure (PSE). The large red-shift of the C=O stretching mode indicates that strong H-bonds in the dimer form appear in the solid state through this group.

## 1. Introduction

The compound 1,2,3-triazole is a privileged structure and one of the most important classes of nitrogen-rich heterocyclic scaffolds. Their derivatives have also shown a profound activity to inhibit cancer cell proliferation and to induce cell cycle arrest and apoptosis [1,2]. The specific behavior of triazoles in biological systems is associated with the presence of three nitrogen atoms in their heterocyclic core, which together with heteroatoms of side substituents and functional groups are capable of binding to the active sites of various enzymes and receptors through weak intermolecular interactions. Due to this ability, triazoles are being widely tested for different biological activities, as well as for the synthesis of hybrid structures with drugs to enhance their binding to cellular receptors [3]. This fact allows the potential use of these compounds in medicinal chemistry [4,5,6,7,8].

Previously, the synthesis of a series of new 1,2,3-triazole derivatives with different substituents at the N2-aryl ring, with the carboxylic (or carboxylate) group, in addition to the pyrrolidine group was reported [3]. Among these derivatives, two of them are now the focus of detailed studies from theoretical and experimental spectroscopy points of view. The aim of this investigation is the study of the fine molecular structure of 2-aryltriazoles, since the molecular architecture in biological active compounds affects its binding force to the active sites of receptors or targeting molecules, and therefore, determines its selectivity and cytotoxicity degree. Therefore, the knowledge of the electronic density distribution in a molecule, the greatness of its charges on the nucleophilic and electrophilic centres and its dipole moment, magnitudes that are calculated in the present work, will be very important to predict its biological effects [1,2] and to design new candidates with improved properties.

The selected molecules were 2-(4-methoxyphenyl)-5-(pyrrolidin-1-yl)-2*H*-1,2,3-triazole-4-carboxylic acid (molecule **1a**) and its anion form (molecule **2a**), Figure 1. These molecules can have special properties due to the liposolubility of the structure provided by the aromatic ring that facilitates cell membrane penetration, and its hydrosolubility provided by the carboxylic group or its anionic form. In addition to these compounds, several derivatives (**1c–e** and **2c–e**) with electron-donating group in the *para*-position at the aryl ring were also optimized. Therefore, the effect of the substituent electronic nature on the structural parameters of the triazole ring was evaluated, which will be useful for further designing new compounds with improved biological activity.

In these molecules, non-bonded interactions with amino acids in any particular biological target are weak, but they build a fine supramolecular structure, which will later affect many properties that will allow their pharmacological application. Therefore, the differences between the geometry and electronic characteristics of the dimers and their separate molecules of thiazole acid and its salt will be reported, and the dependence of these characteristics on the electronic nature of the substituent at the aryl fragment will be clarified.

Vibrational FTIR spectra of related triazole molecules have been reported [9], although not in such detailed form as presented here, nor using accurate scaling procedures. For this purpose, the crystal unit cell in the solid state of **1a** and **2a** was simulated as a dimer form and optimized. Therefore, an improvement in the theoretical spectra was achieved, which facilitated the assignment of the experimental ones. 

## 2. Results and Discussion

One of the goals of the present manuscript was to know the effect of several substituents on the molecular structure and atomic charge distribution of the triazole ring. Its acknowledgement will facilitate the selection of the substituents on the triazole ring for synthesis of new bioactive compounds. In the present paper, a few substituents in the *para*-position of the aryl ring were tested, Figure 1. Of these compounds, 2-aryltriazole acids, **1a** and **1b**, and their corresponding anionic forms were previously synthesized, and their NMR and fluorescent properties studied [3]. The molecular structure and vibrational spectra of **1a** (R = OCH_3_) and its anionic form **2a** were interpreted in detail in the present manuscript, including their dimer forms, using DFT and MP2 quantum chemical calculations.

### 2.1. Molecular Geometries in the Monomer Form of **1a** and **2a**

In the monomer form, four conformers appear stable in **1a** by rotation of both the methyl group around the C_1_-O bond and the −COOH group. The most stable one (conformer *1*) is that plotted in Figure 2 and the calculated values of this conformer were only discussed in the present manuscript. Conformer *2* was obtained by rotation around C_1_−O bond, which was included in Appendix A. The energy difference between both conformers is very small, 0.01 kJ/mol at MP2 level. However, the rotation around the -COOH group leads to a larger difference, 10.6 kJ/mol, for conformers *3* and *4*. The increases in the out-of-planarity of this COOH group with a torsional angle C_8_−C_9_−C_11_=O_12_ of 28.2° contributes to this feature. Due to their lower stability, these conformers were not analyzed in the present work.

Several selected optimized geometrical parameters, namely bond lengths, bond angles and torsional angles calculated using the B3LYP and MP2 methods and the 6-31G(d,p) basis set are collected in Table 1 for **1a** and **2a**, as well as with all the molecules of Figure 1, but only at the MP2 level for simplicity. More geometrical parameters are collected in Appendix A-SUP. The optimized structure in the neutral form **1a** and its anion form **2a** are shown in Figure 2, while those corresponding to molecules **1b** (R = CL) substitution), **1c** (R = OH), **1d** (R = ONH_2_) and **1e** (R = OCF_3_) are included in Figure 3. The label of the atoms is in accordance with that reported in a previous study [3]. A few values of several bond lengths of interest are also included in these figures. In the bottom of each structure (Figure 2) is shown the total energy (*E*) values, which includes the ZPE (zero-point vibrational energy) correction and the Gibbs energy (*G*). In Figure 3, the energy values by MP2 were only included for simplicity.

In general, comparing the geometric parameters of the most stable forms of **1a** and **2a**, it is noted that the change of the –COOH group in **1a** (neutral form) by –COO^−^ in **2a** (anion form) leads to large differences in the whole molecular structure. These differences are noticeable more than if different substituents are inserted in the *para*-position (on C_1_) of the aryl ring, such as in molecules **1b** to **1e**.

In these molecules, both the aryl and triazole rings show a full planar structure, as can be seen from the side views of these figures. However, the pyrrolidine ring is clearly non-planar as was expected. This non-planarity measured in the torsional angle N_14_-C_15_-C_16_-C_17_ is higher in 7.5° (B3LYP) and 17.8˚ (MP2) in the neutral **1a** than in the anionic form **2a**. 

The substituents on the triazole ring are not coplanar with it. Thus, the aryl ring appears noticeably rotated related to the triazole ring, with values of the torsional angle C_5_−C_4_−N_4_−N_10_ remarkably higher in the neutral form **1a** (−14.2° by MP2) than in the anionic form **2a** (−3.0°). B3LYP fails in the calculation of this angle with very low values, because the conjugation of the electron lone pair of the nitrogen atoms with the ring is worse described than by MP2 [10]. The pyrrolidine ring plane is also out of the triazole ring plane with values of the torsional angle C_9_−C_8_−N_14_−C_18_ of 26.4° by MP2 in **1a** and 35.6° in **2a**. B3LYP also fails in the calculation of this angle with lower values. This non-coplanarity of the triazole substituents was also observed in molecules **1b** to **1e** with small differences in the torsional angles, around 2–3°.

The carboxylic group appears remarkably rotated related to the triazole ring, while the coplanar form corresponds to a saddle point. This rotation is especially large in the anionic form **2a** with a value by MP2 of the torsional angle C_8_−C_9_−C=O_12_ of 26.5° vs. 15.0° in **1a**. This feature can be explained by a larger flexibility of the carboxylic oxygens in the **2a** form with a longer C=O_12_ bond length of 1.268 Å vs. 1.225 Å in **1a**. This flexibility is regulated by a strong O_12_···H_18_ H-bond, which is shortened in the anionic form compared to the neutral form by 0.151 Å (Figure 2). This hydrogen bond changes minimally in the triazoles **1b–e**. However, this non-planarity of the structure is expected to be reduced in the solid state because the intermolecular H-bonds and packing forces of the crystal will tend to compress the structure. 

The loss of the H_13_ proton in the anionic form leads to a noticeable effect on the structure of the molecule. It is confirmed by the charge values on the carboxylic group atoms, with larger negative charge localization on the oxygen atoms O_12_ (−0.889*e*) and O_13_ (−0.843*e*), with arrangement of C_11_-O_13_ (1.252 Å) and C_11_-O_12_ (1.268 Å) bonds and an increase in C_9_−C_11_ bond length of up to 1.549 Å and with a higher flexibility of this group (Table 1 and Table 2). This lengthening of the C_9_−C_11_ bond in the anionic form leads to an increment of the double bond character of N_7_=C_8_ and C_9_=N_10_ of the triazole ring and consequently the C_4_−N_4_, N−N and C_8_−N_14_ bonds are lengthened. This feature modifies, in 2°, the bond angles of this triazole ring and, especially, the torsional angles and the non-planarity with its substituents, Table 1. 

In the neutral form of these molecules the highest negative charge corresponds to the hydroxyl oxygen O_13_ (−0.776*e*), Table 2. O_12_ and O_1_ oxygens also have a large negative charge and they can take part in the binding of these molecules with biological targets. The negative charge on the nitrogen atoms N_7_ and N_14_ is lower (−0.188*e* and −0.388*e*, respectively) but it is expected that they can also affect the biological behavior of these molecules. By contrast, the highest positive charge appears on the C_8_ and C_11_ carbon atoms because they are bonded to highly negative charged atoms.

### 2.2. Relationships Established between the Molecular Parameters

Several relationships were established between the optimized geometric parameters and electronic characteristics of the five acids **1a–e** shown in Figure 1. The different *para*-substituents on the aryl ring leads to a noticeable change in the negative NBO natural atomic charge of O1 oxygen, which is larger in compound **1c** (R = OH), and lower in compound **1d** (R = ONH_2_), Table 2. The lowest value corresponds compound **1b** with the chlorine substituent bonded to the aromatic ring. With the exception of acid **1b**, their values appear linear related to the dipole moment of the molecule, Figure 4a. A decrease in the negative charge on O1 leads to an increment in the dipole moment, as is expected due to the large negative charge of the other side of the molecule with the -COOH group. The direction of the dipole moment vector differs in **1b** (R = CL) and **1e** (R = OCF_3_) due to the lack of O1 in **1b** and the three fluorine atoms in **1e**.

With the exception of **1b** (R = CL) and **1e** (R = OCF_3_), a decrease in the negative charge on O1 leads to a slight increment in the intramolecular distance of the most reactive O1 and O12 atoms, Figure 4b. If this distance is increased, it could affect its arrangement and further interaction with the protein cavity amino acids, and therefore, its anticancer activity and selectivity.

A change in the negative charge on O_1_ with the different *para*-substituents on the aryl ring leads to a change in the positive charge on the C_1_ bonded atom. The exception is **1b** with a negative charge on C_1_ due to there being a chlorine atom instead of O_1_. This feature leads to a redistribution of the aryl group charge, which affects its aromaticity and the positive charge value on C_4_. An increment in its positive value appears linear related to a shortening of the C_4_−N_4_ bond length, Figure 5a. Although this change in **1b** (R = CL) and **1e** (R = OCF_3_) looks small, it is enough to give rises to a decrease of 4º in the rotation of the aryl ring plane related to the triazole ring plane, which can be measured through the C_5_-C_4_-N_4_-N_10_ torsional angle. This decrease is lower in the anionic forms.

An increase in the positive charge on C_4_ give rises to an increment in the charge on N_4_ of the triazole ring, which is negative in **1b** and **1e**, and becomes very minimally positive or almost null in the other molecules **1a**, **1c** and **1d**, Figure 5b, and it is also related to the C_4_-N_4_ bond length. Variations in the N_4_ charge appear to have a significant effect on the torsional angles of the triazole substituents. Therefore, an increment in its positive value is linear related to a decrease in the torsional angles C_5_-C_4_-N_4_-N_10_ (Figure 5c), C_4_-N_4_-N_10_-C_9_ (Figure 5d) and N_10_−C_9_−C_11_=O_12_ (Figure 5e), and an increment in the C_9_-C_8_-N_14_-C_18_ (Figure 5f) and N_14_-C_15_-C_16_-C_17_ (Figure 5g). All these results show how the different substituents affect the triazole ring geometry and electronic structure. Although the changes observed appear small, they could be of importance in the sensitivity of these molecules to external stimuli in microenvironments, including their activity to biological targets.

### 2.3. Molecular Geometries in the Dimer Forms

In compounds where a suitable crystal for X-ray analysis is difficult to obtain or it is not available, IR and Raman spectra are used to characterize the compound structure. For this task, it is necessary that the wavenumbers of the calculated structure clearly correspond to those of the experimental ones in the solid state. Otherwise, the optimized structure differs from the experimental one. Therefore, a slight improvement of the theoretical model with the cyclic dimeric form in **1a** and a stacking form in **2a** was carried out. This slight improvement leads to our calculated IR and Raman spectra to appear closer to the experimental ones. It permits us to assert that the synthesized compounds correspond to the calculated ones, and also an accurate assignment of the vibrational bands, especially those corresponding to the triazole ring, which is another objective of the present manuscript.

In the solid state, **1a** is expected to be symmetrically H-bonded in a cyclic dimer form through the –COOH group as in related molecules with the carboxylic group [11]. Thus, its dimeric structure was optimized and plotted in Figure 6. This optimized dimer form predicted for the crystal unit cell was confirmed by comparison of the carboxylic vibrations of their IR and Raman spectra with those obtained experimentally in the solid-state sample. Both molecules of the dimer are almost planar, and they are H-bonded with the same bond length, 1.646 Å by B3LYP. This H-bond value is slightly longer than that reported in the benzoic acid (BA) dimer, 1.616 Ǻ [11], which can be explained by the slight decrease in the negative charge on =O_12_ in **1a** as compared to BA. A longer C=O bond (1.241 Ǻ vs. 1.237 Ǻ in BA) and, consequently, a shorter C_9_-C_11_ bond length (1.466 Ǻ vs. 1.487 Ǻ in BA) is calculated. As expected, a lengthening of the acceptor C=O bond (1.225 Å in monomer vs. 1.241 Å in dimer) and a shortening of the donor C-OH bond (1.357 Å in monomer vs. 1.320 Å in dimer) is observed on dimer H-bond formation. This shortening in the C-OH bond is proportional to the intermolecular H-bond length [12]. 

The anion **2a** cannot be in cyclic dimer form in the crystal through an O-H···O bond as in **1a**. Thus, several stacking forms between two molecules were optimized as a simplified model, with interaction of both -COO and NNN moieties as well as π-π interactions. Figure 7 shows the two best optimum stable forms calculated with the M06-2X method, since B3LYP fails in the stacking interactions [13]. The most stable one corresponds to form *I*, and only for this form were the vibrational spectra studied. A comparison of its theoretical scaled vibrational spectra with the corresponding experimental one, especially in the stretching vibrations of the −COO and NNN modes, appears to confirm this stacking form or a similar one.

*Form I* appears stabilized by several weak H-bonds/interactions through the oxygen atoms and the out-of-plane methyl hydrogens of the pyrrolidine ring. An increment in the twist of the –COO group is observed to facilitate these intermolecular H-bonds/interactions. A high planarity appears between both monomers of this dimer form, and because of that, this structure is expected in the crystal unit cell. *Form II* is slightly less stable than *form I*, and it is also stabilized by weak C-H···O H-bonds/interactions between the –COO group and the CH_2_-hydrogens of the pyrrolidine ring. In addition, a weak C-H···N H-bond/interaction is observed. However, the planarity between both monomers is remarkably reduced in this dimer and therefore, it is expected that the crystal packing forces modify this structure. Other dimer forms were attempted to be optimized with several H-bonds/interactions through the nitrogen atoms, but they were not stable.

The interaction energies were calculated in **1a** and **2a** dimers. Its calculation is described in detail in the Appendix A. The deformation energy *E*^def^ in the **1a** dimers, 18.6 kJ/mol, is slightly lower than in the most stable form of **2a**, 21.8 kJ/mol. This can be due to the stacking interaction, which, although weak in the **2a** dimer, affects a larger number of atoms than in the **1a** dimer. This deformation energy has a similar value in the *I* and *II* forms of **2a**. The CP corrected interaction energy of the $\Delta E_{}^{CP}$ is −82.8 kJ/mol in **1a**, in accordance with that reported in the cyclic dimer of benzoic acid, −81.7 kJ/mol [11].

### 2.4. Molecular Properties

Several thermodynamic parameters, rotational constants and dipole moments were also calculated for dimer forms and compared to those for individual molecules (monomers). These data obtained in the global minimum were included in Table 3. In general, the computed values by B3LYP are close to those by M06-2X, with small differences. In the dimer form, the rotational constants values are remarkably reduced, around five times lower than those in the monomer form. By contrast, the values of C_v_ and entropy (S) are twice as higher in the dimer form than in its monomer. The values of these parameters appear similar to the other molecules under study. 

However, the dipole moment value in the anion form **2a** is remarkably higher, 10 times, than in its neutral form **1a**. This is the main difference among these molecules. This feature is in accordance with a higher solubility in water of **2a**. By contrast, the value in **1a** is too low and it has no water solubility. In the dimer forms, the dipole moment value is lower in **1a** than in its monomer, but in the **2a** anion, it is remarkably lower in its dimer in accordance with the arrangement of the –COO groups.

### 2.5. Scaling the Wavenumbers

Because theoretical methods do not adequately reproduce all the experimental patterns of wavenumbers with enough accuracy, the use of scaling procedures is necessary to improve the results remarkably [14,15]. The linear scaling equation procedure [16], using one (LSE) or two equations (TLSE) (for high and low wavenumbers) represents a compromise between accuracy and simplicity, and therefore, they were the main procedures used to assign the experimental bands. In addition, the polynomic scaling equation procedure (PSE) was also used. These procedures use the equations calculated in simpler building molecules, which, in the present study, the results used were those of the benzene molecule at the same level of theory. The calculated wavenumber by the theoretical method is represented by ν^cal^, and the scaled wavenumber by v^scal^. The scaled values can be obtained by the following procedures: (a) the LSE equation [16], which is:ν^scal^ = 22.1 + 0.9543 · ν^cal^ at B3LYP/6-31G(d,p) level

(b) by the TLSE procedure, which are:ν^scal^ = 29.7 + 0.9509 · ν^cal^ at B3LYP/6-31G(d,p) level for the 1000–3700 cm^−1^ range
ν^scal^ = −16.0 + 1.0009 · ν^cal^ at B3LYP/6-31G(d,p) level for the 0–1000 cm^−1^ range

(c) Finally, by the PSE procedure, which are:ν^scal^ = −4.2 + 0.9909 · ν^cal^ − 0.00000929· (ν^cal^ )^2^ at B3LYP/6-31G(d,p) level
ν^scal^ = 6.5 + 0.9694 · ν^cal^ − 0.00000612· (ν^cal^ )^2^ at M06-2X/6-31G(d,p) level 

The equation corresponding to the M06-2X/6-31G(d,p) level was only used for the dimer form of **2a**, because of the lower accuracy [16] of this M06-2X method than B3LYP in the scaled wavenumbers.

### 2.6. Vibrational Analysis of **1a** and **2a**

All the calculated wavenumbers in the most stable form *I* are collected in Appendix A . A short resume of the most important values is shown in Table 4 and Table 5. Because the geometrical values and vibrational wavenumbers of form *2* are almost the same as those of the form *1*, they were not included in the Tables. Only the wavenumbers with high IR or Raman intensity, or those characteristics of the molecular structure were included. The scaled wavenumbers by two methods, the relative (%) computed IR and Raman intensities, the experimental values observed in the spectra and the main characterization of the vibrations with their % contribution of the different modes to a computed value (PEDs) are also included in the Tables. Contributions in general lower than 10% are not included. The relative intensities are obtained by normalizing each calculated value to the intensity of the strongest one. 

The scaled IR and Raman spectra were mainly carried out using the TLSE and PSE scaling procedures. The LSE procedure is the worst, while the PSE procedure leads to the best results, with errors in general lower than 3%. The scaled wavenumbers are slightly worse using the LSE and TLSE procedures than the PSE. Thus, all the scaled spectra shown in the present manuscript were performed with this PSE procedure, as well as its discussion.

A comparison of the whole FTIR experimental spectra of **1a** and **2a** with those corresponding to the theoretical scaled spectra by the PSE procedure in the monomer form were plotted in Appendix A, while the comparison with the Raman values is shown in Appendix A. For a better analysis and comparison of the different experimental and scaled vibrational wavenumbers of these figures, the spectra are divided into three regions, such as: 3700–2700 cm^−1^, 1800–1000 cm^−1^ and 1000–0 cm^−1^ (or 1000–600 cm^−1^). The IR spectra of these figures are shown in Figure 8, Figure 9 and Figure 10, while for simplicity, the Raman spectra are included as Appendix A. The assignment of the most intense and characteristic IR wavenumbers is included in these figures.

In a general comparison of the IR spectra in Appendix A, the following is observed: (i) A large difference between the spectra of **1a** and **2a**. This difference is in agreement with a significant change in the geometric parameters and charges between both molecules. Although acid and salt have almost the same chemical structure, they can differ significantly in their biological effect, since they differ greatly in electronic and geometric characteristics both in the form of individual molecules and in the form of dimers. This can be manifested in their affinity to a specific receptor site and in their value of the effect and selectivity. (ii) A noticeable accordance between the scaled wavenumbers in the monomer form with the experimental ones, with only a few significant differences. (iii) The scaled spectra in the dimer or stacking forms reduce the differences between theory and the experiment.

It is noted that most of the modes in the compounds under study appear in the expected ranges. Due to this feature and because the difference in the observed and scaled values of most of the fundamentals is very small, the assignments in general could be considered correct. This assignment of the vibrational bands was carried out through a detailed comparison of the experimental bands with the scaled spectra. This assignment was discussed under the following sections: (i) The COOH and COO group modes, (ii) the OH group modes, (iii) the triazole ring modes, (iv) the phenyl ring modes, and (v) the methoxy O-CH_3_ modes. The discussion was carried out mainly focused on (i) and (ii) sections because they involved the most reactive groups.

#### 2.6.1. The Carboxylic COOH Group Modes in Molecule **1a**

The displacement vectors for the characterization of these modes are similar for the monomer and dimer forms, although for each dimer vibration two wavenumbers appear, one corresponds to the in-phase mode (Raman active) and another one to the out-of-phase mode (IR active). Table 5 collects a resume of the calculated (scaled) wavenumbers, together with the experimental ones and the main characterization with the %PED. The full Table is included as Appendix A. The theoretical values mainly correspond to the monomer form of the molecules, and only when noticeable differences appear with the dimer they are included in this table.
ijms-24-14001-t005_Table 5Table 5Calculated harmonic wavenumbers (ν, cm^−1^), relative infrared intensities (A, %), relative Raman intensities (S, %) and scaled values (ν, cm^−1^) in the COOH (**1a**) and COO (**2a**) groups.GroupMode
ASTLSEPSEIRRamanCharacterizationν^cal^ν^scal^ν^scal^COOHν(O-H) δ(O-H)γ(O-H)376231611269591151001311170213607303512365763592303512385783596.1 w3491.0 w1243.1 vs3504.2 vw1243.0 w570.9 wν(O-H) (100)Dimer: ν(O-H) out-of-phaseδ(O-H) (52) + ν(CN)(25) + γ_as_(CH) pyrrol (16)γ(O-H) (81)ν(C=O)1743170280007016871648169516551675.1 vs1643.2 mDimer: ν(C=O) out-of-phase Dimer: ν(C=O) in-phaseν(C-O)1140111929300311141094111310931122.5 m1093.6 m1120.6 w1096.5 wν_as_(COO) (33) + ν_s_(NNN)(31) + 15,δ(CH)(28)ν_s_(COO) (45) + δ(NNN) (38)δ(C=O)7147967779150008699782763699780761697.2 w779.2 vs696.3 w768.0 mδ(COOH) (46) + γ(triazole) (38)Dimer: δ(COOH) out-of-phase + ν(CC)Dimer: δ(COOH) in-phaseγ(C=O)72191706705707.8 vvw705.9 wγ(COOH) (62) + γ(NC_8_C) (21) + γ(CN_14_) (16)COOν_as_ν_s_δ_as_γ_s_175913377958088335251241117021301786798171013047787901588.3 vs1299.9 m1629.7 vw1307.6 vw774.4 w783.1 vwν_as_(COO) (96)ν_s_(COO)(34) + ν(triazole)(32) + γ(CH)pyrrol(18)δ_as_(COO) (58) + δ_as_(C-H) in pyrrolidine (15)γ_s_(CCOO) (55) + γ(C_8_C) (30) + 6a, δ(CC) (27)Notation used for experimental bands: vs = very strong, s = strong, m = medium, w = weak band, vw = very weak.


The experimental IR spectra of Figure 10 are characterized by a broad and complex spectral structure around 2850 cm^−1^, which is characteristic of the carboxylic acid association by hydrogen bonding. Only the internal modes of the COOH and COO groups were discussed under the following sections: 

*The C=O modes*: The C=O stretching was calculated in the monomer form with very strong IR intensity and scaled by PSE at 1745 cm^−1^. However, this value noticeably differs (70–100 cm^−1^) from that observed at lower values in the experimental IR and Raman spectra, which is due to intermolecular H-bonds present in the solid state through the carboxylic –COOH group. In our dimer simulation, this mode was scaled at 1695 cm^−1^ with very strong IR intensity also. This value is slightly higher than 1675.1 cm^−1^ found experimentally. This feature indicates an intermolecular H-bond slightly stronger in our calculated dimer form than in the experimental sample of **1a**. This is expected and in accordance with additional intermolecular interactions with other dimer forms present in the stacking form in the solid-state crystal that slightly lengthens the C=O bond length. 

Another ν(C=O) stretching vibration is predicted at 1655 cm^−1^ in our optimized form of **1a**, but with the displacement vectors of the stretch motion appearing in-phase between both monomer forms of the dimer. This type of motion leads to an active Raman band with very strong intensity, but inactive in IR. Our predicted wavenumber in Raman is in good accordance with the experimental Raman line at 1643.2 cm^−1^. In the solid state of BA, this mode has been reported experimentally [11] at 1693 cm^−1^ (IR) and 1635 cm^−1^ (Raman), in accordance with our results.

The C=O in-plane bending appears strongly coupled with the δ(C-OH) mode, and therefore, can be better denoted as δ(COOH) in Table 5. It is predicted in the monomer form at 699 cm^−1^. However, in the dimer form it is scaled at 780 cm^−1^ (IR, out-of-phase motion) with medium IR intensity and at 761 cm^−1^ (Raman, in-phase motion) with a weak value, in excellent accordance with the very strong experimental IR band at 779.2 cm^−1^ and to the medium intensity Raman line at 768.0 cm^−1^. The discussion of the out-of-plane modes is included in the Appendix A.

*The C-O_13_ modes*: The displacement vectors correspond to the COO group instead of an isolated C-O bond and is therefore characterized as ν_as_(COO) in Table 5. In the monomer form, its stretching mode appeared scaled with strong IR intensity at 1113 cm^−1^ and it could be related to the experimental bands at 1122.5 cm^−1^ (IR) and 1120.6 cm^−1^ (Raman). In the dimer form, they were predicted with very weak IR and Raman intensities, and therefore they were not detected in the spectra. This stretching mode in **1a** appears noticeable coupled with ν_s_(NNN) and δ(C-H) modes, and thus, their experimental wavenumbers differ from those assigned to this mode in BA at 1347 cm^−1^ (IR) [18,19]. 

The ν_s_(COO) mode appears strongly coupled with the ν(NNN) stretching of the triazole ring which complicates its identification. It was characterized in the scaled wavenumber of the monomer form at 1093 cm^−1^ appearing with medium-strong IR intensity, and thus, was well related to the IR band with medium intensity at 1093.6 cm^−1^. In Raman, it was predicted with weak intensity and was related to the Raman line at 1096.5 cm^−1^. In the dimer form, this mode was predicted with very weak and almost null intensity, and therefore, was not related to an experimental band. 

*The O_13_-H modes*: The stretching ν(O-H) of free hydroxyl groups (monomer form) appears scaled at 3592 cm^−1^ with medium IR and Raman intensity and were related to the weak experimental IR bands at 3596.1 and 3491.0 cm^−1^, and to the Raman line at 3504.2 cm^−1^. This means that free COOH groups appear in **1a**, in accordance with the experimental IR band reported at 3553.3 cm^−1^ in *p*-methoxybenzoic acid [19]. These features are also in accordance with those reported in BA [11], where the monomer form is calculated by B3LYP/6-31G(d,p) at 3763 cm^−1^ (scaled by PSE at 3593 cm^−1^), at almost the same wavenumber as our value in **1a** at 3592 cm^−1^. However, a medium intensity band at 3567 cm^−1^ has been found [20,21] in the experimental IR spectrum of the BA molecule, which appears at a slightly lower wavenumber than in **1a** at 3596.1 cm^−1^, perhaps due to a lengthening of the OH bond by weak interactions in the solid state. A weak and broad IR band is also detected in **1a** at 3491.0 cm^−1^ (at 3504.2 cm^−1^ in Raman) which can only be due to the stretching of free O-H groups weakly intermolecularly H-bonded to other molecules. 

In H-bonded hydroxyl groups (cyclic dimer form through this group), the OH stretching wavenumber is red shifted (scaled) at 3035 cm^−1^ (IR) and 2943 cm^−1^ (Raman) and is predicted with the strongest intensity of the spectra. This highest intensity in the dimer form appears in accordance with that calculated in the dimer of BA [11], and to that reported in carboxylic acids [22,23], where it is usually identified by a broad stretching band near 3000 cm^−1^. However, bands with strong or very strong intensity have not been observed in the stretching region of the experimental IR and Raman spectra. Perhaps due to this, they are included in the very broad band with weak-medium intensity observed in the IR spectrum at 3054.1 cm^−1^, closely to our calculations in the dimer form. This feature, together with the red shift of the ν(C=O) wavenumber in the dimer form and the experimental bands assigned to the δ(C=O) mode, indicates that in the solid state most of the molecules appear H-bonded in cyclic dimer forms through the COOH group but there are also molecules that remain free of H-bonds. 

The in-plane bending δ(O-H) appears strongly coupled with CN and C-H modes, as well as with other modes. In the monomer form, it appears scaled at 1238 cm^−1^, in accordance with the experimental IR band observed at 1243.1 cm^−1^ and the Raman line at 1243.0 cm^−1^. In BA, it was calculated at 1218 cm^−1^ (scaled at 1189 cm^−1^), slightly lower than our calculations and to the experimental IR value [20,21] at 1169 cm^−1^. In the dimer form of **1a**, this mode appears spread out in many calculated wavenumbers, especially at 1540, 1454 and 1371 cm^−1^. Because this mode does not represent the main contribution in the calculated wavenumber, their values were not included in Table 5. 

The out-of-plane bending γ(O-H) mode appears clearly characterized as an almost pure mode (81% PED) and scaled at 578 cm^−1^, in accordance with the weak Raman line at 570.9 cm^−1^. In the dimer form, it was scaled at 998 and 942 cm^−1^ but with almost null intensity. Thus, only the very weak Raman line at 932.5 cm^−1^ could be tentatively assigned to this mode. In BA, it was reported at 628 cm^−1^ (in free O-H) and at 960 cm^−1^ (in O-H bonded) [11,20,21] in accordance to our results, as well as in *p*-methoxybenzoic acid [18], which was observed at 546 cm^−1^. 

#### 2.6.2. The Carboxylate COO Group Modes in **2a**


The ν_as_(COO) stretching mode was predicted (scaled) with very strong IR intensity in the monomer form at 1710 cm^−1^, but in the experimental IR spectrum bands were not detected in the 1590–2200 cm^−1^ range, and the most closely band appears at 1588.3 cm^−1^ with very strong intensity, which was related to this mode. This large red shift to lower wavenumbers in the experimental spectra can be interpreted because of strong intermolecular interactions of these molecules through the COO group that lengthened the CO bond. This feature is in accordance with the strong IR absorption near 1600–1560 cm^−1^ reported for the carboxylate group (COO^−^) and corresponding to asymmetric stretching vibrations of solid state samples [22,23]. 

Because planar structures cannot be formed in **2a** molecules, stacking forms were optimized at the M06-2X/6-31G(d,p) level, which were stabilized by several interactions/H-bonds of this COO group, Figure 7. These interactions in the stacking form shift the scaled wavenumbers of this mode, but not enough, and they are also far away from the experimental ones. This feature indicates that the packing crystal forces in the solid state are stronger than those in our simplified optimized model, with a shortening in the distance between planes that increase the COO group interactions and a lengthening of their CO bond lengths. Under a stronger packing, the wavenumbers will be closer to the experimental ones.

The symmetric ν_s_(COO) stretching mode appears spread out and strongly coupled with other modes. The highest contribution was determined in the scaled wavenumber at 1304 cm^−1^ with medium-strong IR intensity, in accordance with the IR band observed with medium intensity at 1299.9 cm^−1^, and the very weak shoulder at 1348 cm^−1^. However, our values also appear in strong disagreement with the IR absorption near 1420–1400 cm^−1^ reported for this symmetric stretching in solid state samples of related compounds [22,23]. 

#### 2.6.3. The Triazole Ring Modes

*The NNN modes:* The ν_s_(NNN) stretching appears strongly coupled with the ν(C_4_-N) mode as well as with other ring modes and it is scaled at 1386 cm^−1^ in **1a**. Because of the very weak IR intensity predicted for this mode, it could not be related to an observed band in the experimental spectrum. However, it was predicted with strong Raman intensity, and thus, was related to the strong line at 1383.8 cm^−1^. A similar wavenumber was calculated in the dimer form of **1a** because of the small effect of the dimer structure on the triazole ring. A large contribution of this mode was also observed in the scaled wavenumber at 1372 cm^−1^, whose major contribution corresponds to the C_4_-N stretching.

In **2a**, this mode was predicted with weak-medium IR intensity at 1356 cm^−1^ and almost null Raman intensity, and therefore it was well related to the experimental IR band with medium intensity at 1346.2 cm^−1^ and to the weak Raman line at 1356.8 cm^−1^. In the stacking form, this mode is slightly red shifted because of the weak π-π interaction of this triazole ring with the COO group. This feature confirms the weak effect on the triazole ring of our stacking optimized structure. Large contributions of this mode appear in the scaled wavenumbers at 1399 and 1319 cm^−1^ that were assigned to the C4-N stretching as the main contribution.

The C_8_-N_14_ modes: The stretching mode is predicted with the highest IR intensity and medium Raman activity in 1a is in accordance with the very strong band observed in the IR spectrum and to the medium intensity line in Raman. It was scaled at 1562 cm^−1^ and well related to the IR band at 1565.1 cm^−1^ and Raman line at 1561.2 cm^−1^. Our scaled wavenumber remains almost unchanged in its dimer form. In 2a, it is predicted with strong IR intensity at 1549 cm^−1^ but the closest experimental IR band appears at 1533.3 cm^−1^ and with weak intensity. This discrepancy can be due to the stacking interactions of the solid state. Thus, in the stacking form of 2a it was predicted with strong intensity at 1587 and 1596 cm^−1^ and they can be inside of the very strong IR band at 1588.3 cm^−1^.

#### 2.6.4. The Aryl Ring Modes

The assignments for several aromatic ring modes are obvious and require no further discussion, therefore the attention was focused only on some important modes here. To avoid lengthening the manuscript, this discussion is included in the Appendix A. The assignments of the ring modes followed the Varsanyi notation [17] for a 1,4-disubstituted benzene. 

#### 2.6.5. The Methoxy OCH_3_ Modes

In general, the calculated wavenumbers of the methyl group, namely ν_as_, ν_s_, δ_as_, δ_s_, γ_as_ and γ_s_ agree well with the experimental values. The stretching modes are calculated as pure modes (100% PED) and with weak IR intensity, as well as the antisymmetric in-plane deformations. The symmetric in-plane mode appears as almost pure (87% PED), as well as the γ_as_ modes. However, the symmetric out-of-plane mode appears spread out in many calculated vibrations. 

The C_1_-O_1_ stretching is predicted strongly coupled with phenyl ring modes. By the displacement vectors of the phenyl atoms in **2a**, it was assigned as mode **7a**. It is predicted at 1266 cm^−1^ in **1a** and at 1245 cm^−1^ in **2a** with strong–very strong IR intensity in accordance to strong IR band observed at 1274.9 cm^−1^ and to the very strong band at 1247.9 cm^−1^, respectively. This mode was predicted in Raman weak–very weak intensity.

## 3. Materials and Methods

Triazoles **1a** and **2a** were obtained by alkaline hydrolysis of 2-(4-methoxyphenyl)-5-(pyrrolidin-1-yl)-2*H*-1,2,3-triazole-4-carbonitrile according to previously developed procedures [24,25]. The IR spectra in the powder form were recorded in the 400–4000 cm^–1^ range on a Brüker IFS-66 FTIR spectrometer equipped with a Globar source, Ge/KBr beam splitter and a TGS detector. For the spectrum acquisition, 50 interferograms were collected. The Raman spectrum was registered in the 50–4000 cm^−1^ range on a Brüker IFS 66 optical bench with an FRA106 Raman module attachment interfaced to a microcomputer. The sample was mounted in the sample illuminator using an optical mount without sample pre-treatment. A Nd:YAG laser at 1064 nm was utilized as the exciting source. The laser power was set at 250 mW and the spectrum was recorded over 1000 scans at room temperature. 

### Computational Details 

Density Functional methods (DFT) [26] were mainly used for the calculations, which provide adequate compromise between the computer time and power required for the computations and the desired chemical accuracy of the results. In biomolecules, DFT calculations have provided results which are quantitatively in good accordance with those raised at MP2 level [27,28], and even better for the vibrational wavenumber calculations [14]. The B3LYP/6-31G(d,p), M06-2X/6-31G(d,p) and MP2/6-31G(d,p) theoretical levels were used for geometry optimizations and NBO atomic charges computations [29], while the B3LYP/6-31G(d,p) was mainly used [30,31] in the calculation of the harmonic IR and Raman vibrational wavenumbers. All optimized structures show only positive harmonic vibrations (local energy minima). B3LYP is one of the most cost-effective DFT methods [16] and it has been used satisfactorily in many studies of biomolecules [32,33,34] and in the drug design field [35,36]. The M06-2X method was used to optimize the dimer structures in stacking form of **2a**, because it appears as one of the best options among the meta-generalized gradient functionals for analyzing dispersion–bound systems and it shows good results in non-covalent interactions with broad applicability in chemistry [37,38]. The MP2 method was also used to confirm the stability of all optimized structures. The 6-31G(d,p) basis set was mainly used in all calculations since it appears as the most cost-effective one. All these methods and basis set are implemented in the GAUSSIAN-16 program package [39]. The UNIX version with standard parameters of this package was running in the Brigit computer of the University Complutense of Madrid. Berny optimization under the TIGHT convergence criterion was used. 

## 4. Summary and Conclusions

A set of 1,2,3-triazole derivatives with possible anticancer activity were analyzed in detail, from the structural, electronic and spectroscopy points of view, especially the two with the methoxy substituent on the C_1_ aromatic ring. The most important findings of this study were the following:(1)By rotation on the C-O_1_ bond length, the conformers differ less than 1 kJ/mol, while by rotation on the C_9_-C_11_ bond it is about 10 kJ/mol. This feature means a large flexibility of the substituents bonded to the triazole ring and variability of its spatial arrangements.(2)The ionization of neutral carboxylic acid and formation of the anionic form demonstrated a larger effect on the triazole ring structure and its charges than the electronic nature of different substituents on the aryl ring.(3)The effect of five substituents in the *para*-position of the aryl ring on the molecular structure of the triazole and on its atomic charge distribution was determined and several relationships were established.(4)The FT-IR and FT-Raman spectra in the solid state of **1a** and **2a** were recorded, and an accurate assignment of all bands observed was carried out for the first time. For this task, the calculated wavenumbers were improved by two main scaling procedures, leading the PSE to the best results, with errors less than 3%.(5)The scaled wavenumbers in the acid **1a** dimer and stacking forms of triazole **2a** were in better accordance with the experimental bands than those with the monomer, which confirms our simplified optimized system for the crystal unit cell of the solid state.(6)The large red shift of the ν(C=O) mode to 1675.1 cm^−1^ indicates that strong H-bonds in the dimer form appear in the solid state through this group. These features mean that in the solid state, free and H-bonded COOH groups exist simultaneously.(7)A comparison of the scaled and experimental wavenumbers confirms these features for the solid state, with free and H-bonded COOH groups in **1a**, and stacking forms in **2a**.

The structural and spectroscopic characterization of two triazole derivatives with possible anticancer activity as well as the relationships established with other derivatives could manage the selection of substituents on the triazole ring for the design of new types of antitumor compounds.

## Figures and Tables

**Figure 1 ijms-24-14001-f001:**
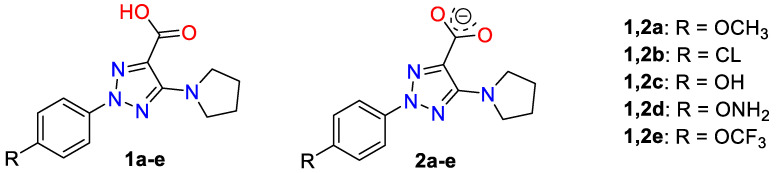
Structures of the triazoles under investigation with their notation.

**Figure 2 ijms-24-14001-f002:**
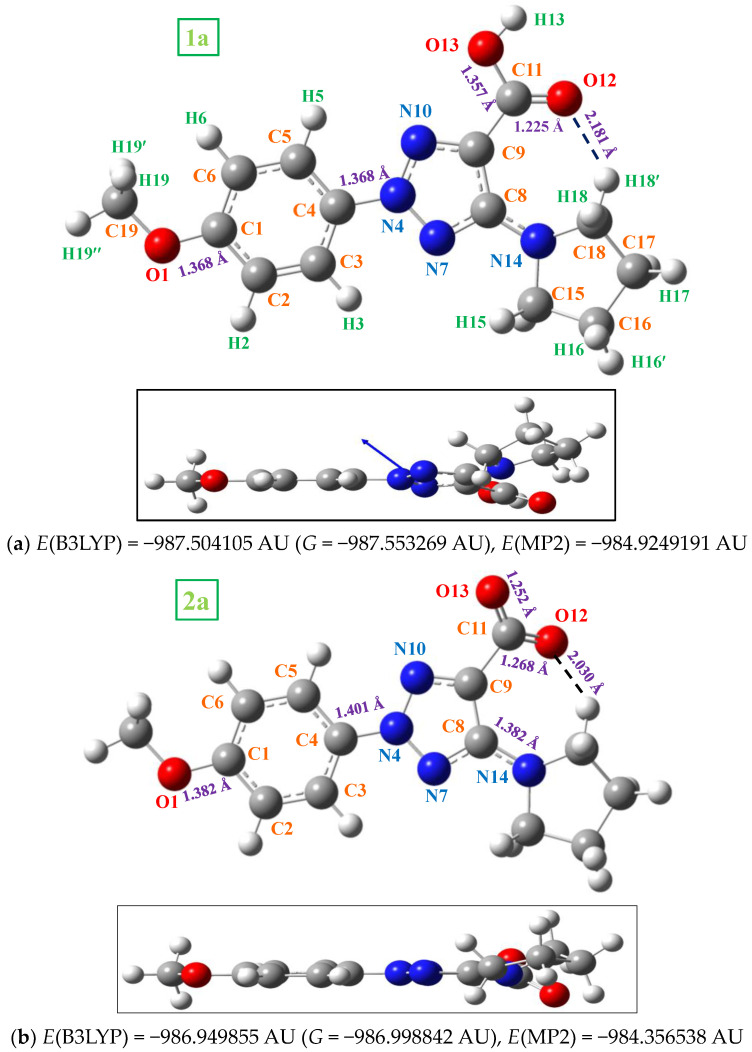
Labeling of the atoms and plot of the structure of: (**a**) 2-(4-methoxyphenyl)-5-(pyrrolidin-1-yl)-2*H*-1,2,3-triazole-4-carboxylic acid (in short **1a**). (**b**) 2-(4-methoxyphenyl)-5-(pyrrolidin-1-yl)-2*H*-1,2,3-triazole-4-carboxylate anion (in short **2a**), with front and lateral view forms. Several bond lengths and intramolecular H-bond values of interest calculated at the MP2/6-31G(d,p) level are included in the figure. The total energy of the system (*E*) including zero-point correction, the Gibbs free energy (*G*) with the B3LYP method and the total energy by MP2 are also included. 1 AU = 2625.5 kJ/mol.

**Figure 3 ijms-24-14001-f003:**
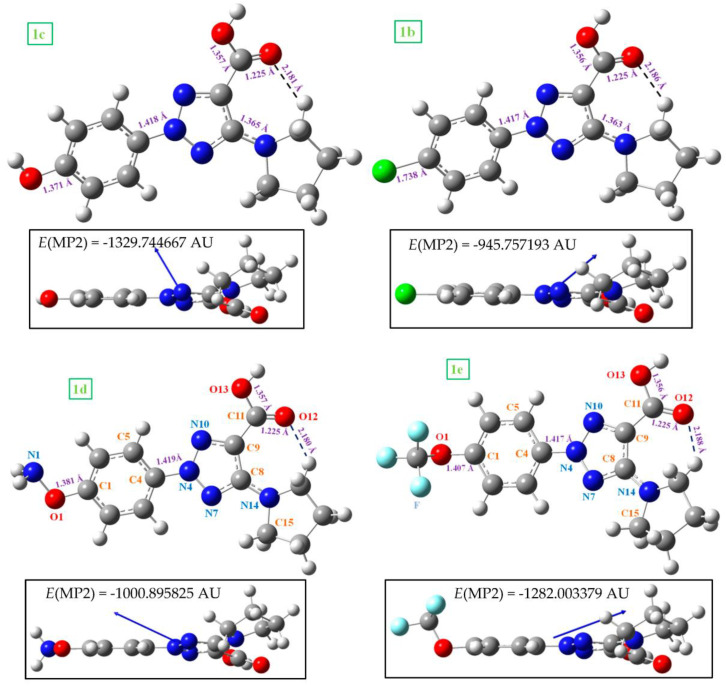
Two views of the optimized structure in the neutral form of **1b** (with chlorine substitution), **1c** (with hydroxyl substitution), **1d** (with the oxo-amino group), and **1e** (with the oxo-trifluoromethyl group). The MP2 energy value was included at the bottom of each figure. The blue arrow corresponds to the dipole moment vector.

**Figure 4 ijms-24-14001-f004:**
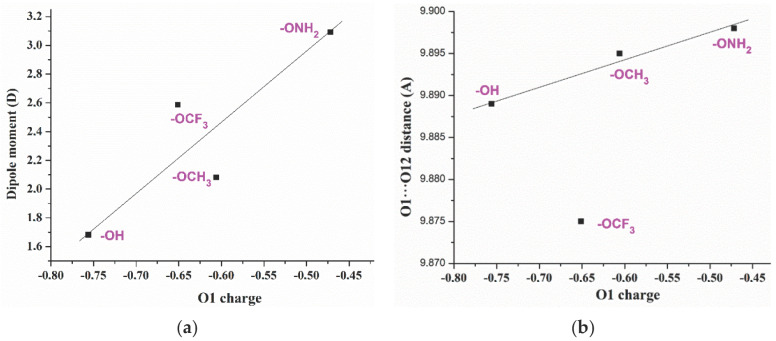
Relationships established between the atomic charge on O_1_ with: (**a**) the dipole moment. (**b**) The intramolecular O_1_···O_12_ distance.

**Figure 5 ijms-24-14001-f005:**
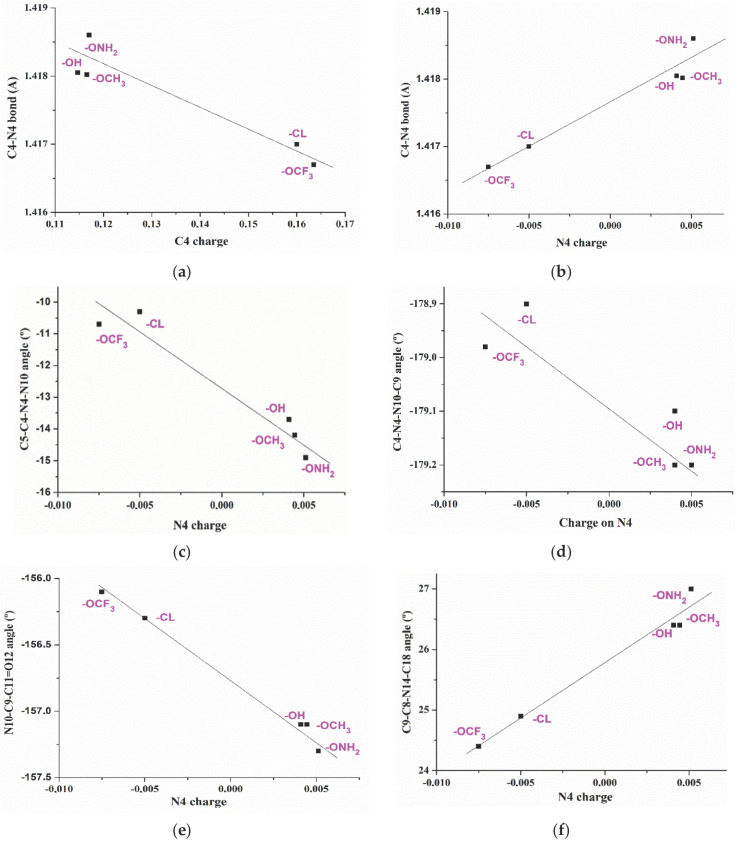
Relationships established between the atomic charges on C_4_ and N_4_ with the geometric parameters involved in the triazole ring structure. (**a**) Relationship between the C_4_-N_4_ bond length and the NBO atomic charge on C_4_. (**b**) Relationship between the C_4_-N_4_ bond length and the NBO atomic charge on N_4_. (**c**–**g**) Relationship between several torsional angles and the NBO atomic charge on N_4_.

**Figure 6 ijms-24-14001-f006:**
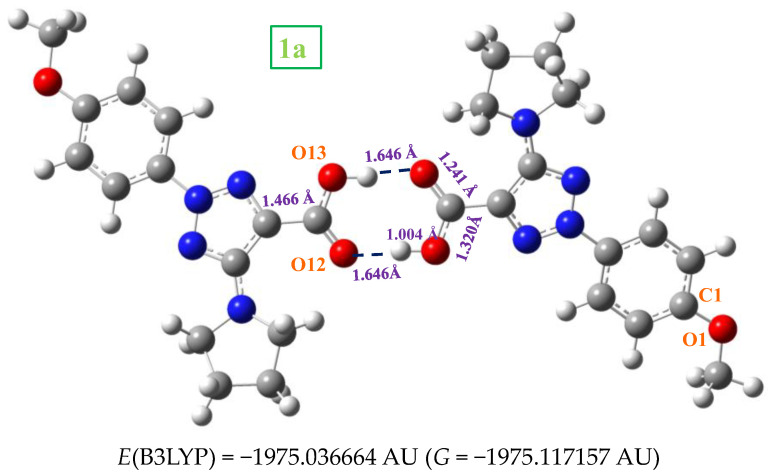
Optimized cyclic dimer form of **1a**. Several bond lengths and H-bond values of interest calculated at the B3LYP/6-31G(d,p) level are included in the figure.

**Figure 7 ijms-24-14001-f007:**
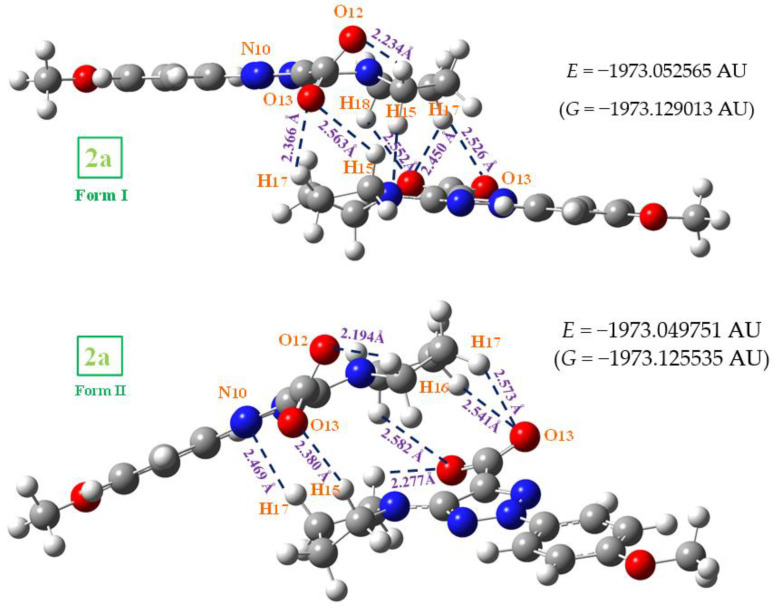
The two optimized dimer forms of **2a**. Several bond lengths and intra- and intermolecular H-bond values of interest calculated at the M06-2X/6-31G(d,p) level are included in the figure.

**Figure 8 ijms-24-14001-f008:**
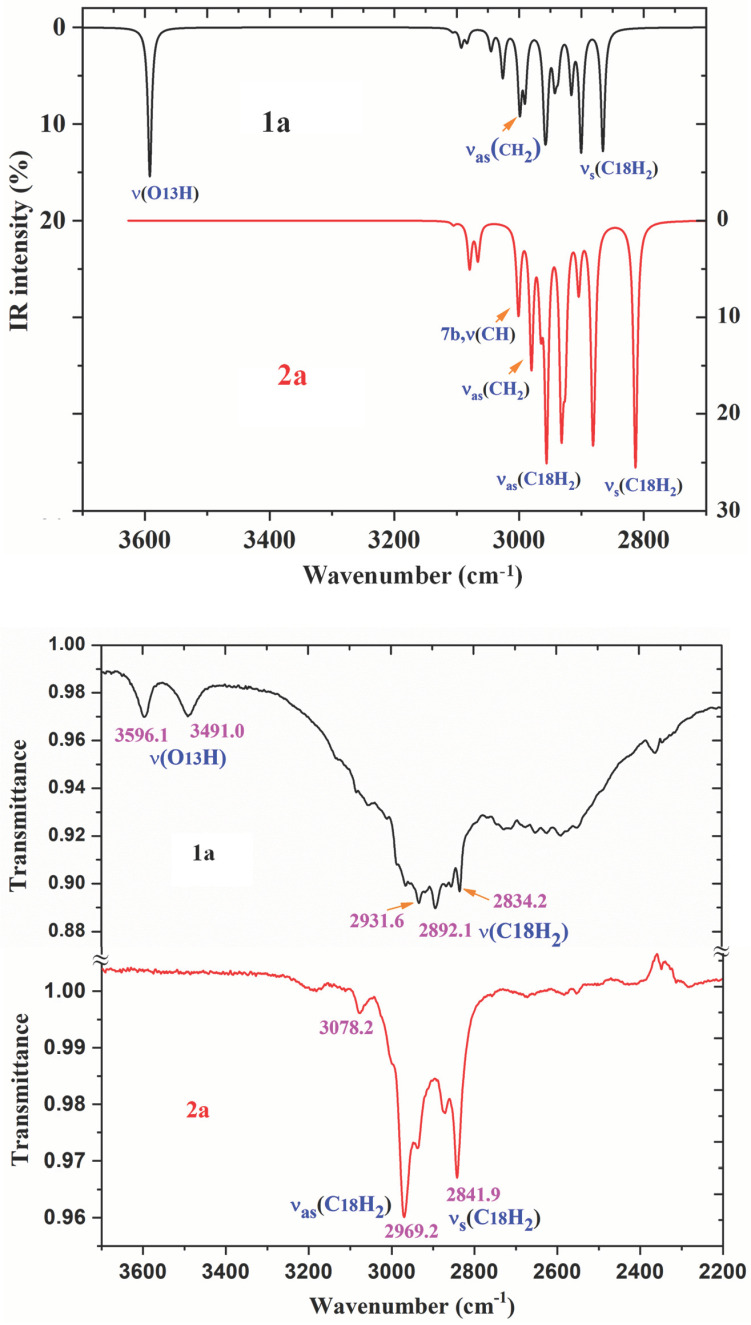
Comparison of the scaled IR spectra of **1a** and **2a** molecules by the PSE procedure in the 3700–2700 cm^−1^ range with the experimental ones in the 3700–2200 cm^−1^ range.

**Figure 9 ijms-24-14001-f009:**
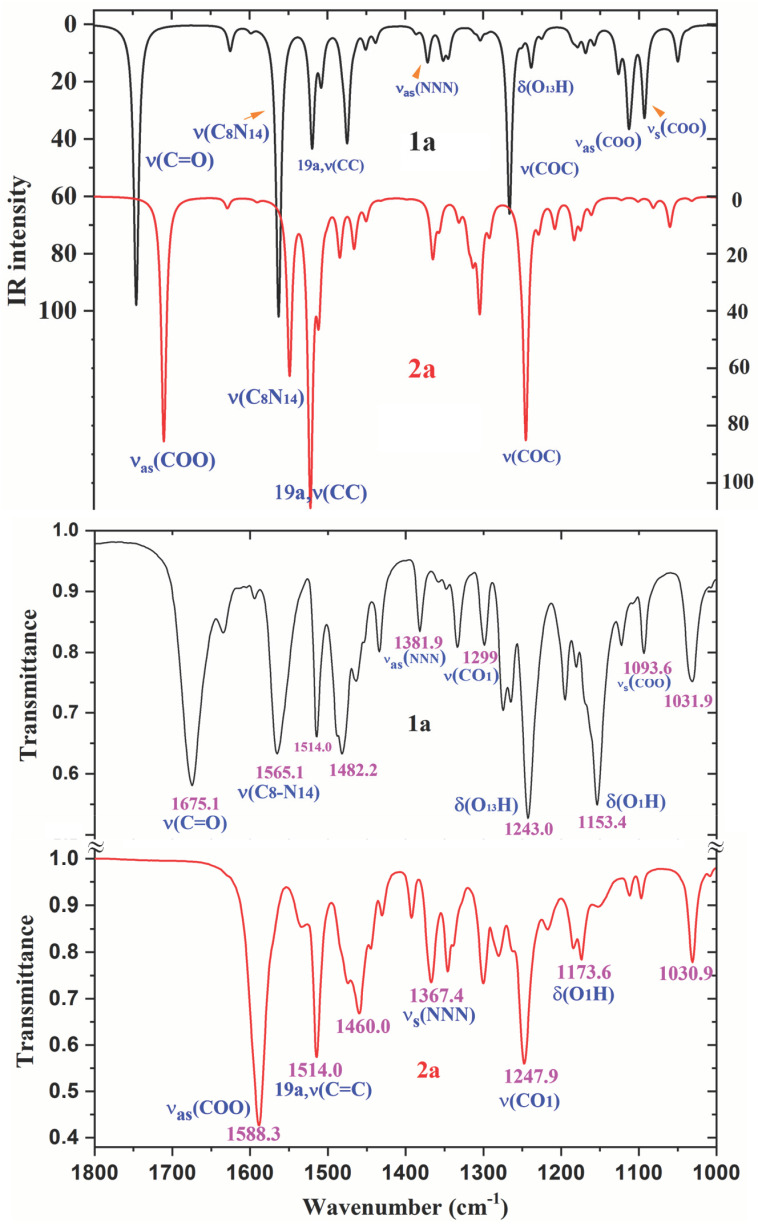
Comparison of the scaled IR spectra of **1a** and **2a** molecules by the PSE procedure with the experimental ones in the 1800–1000 cm^−1^ range.

**Figure 10 ijms-24-14001-f010:**
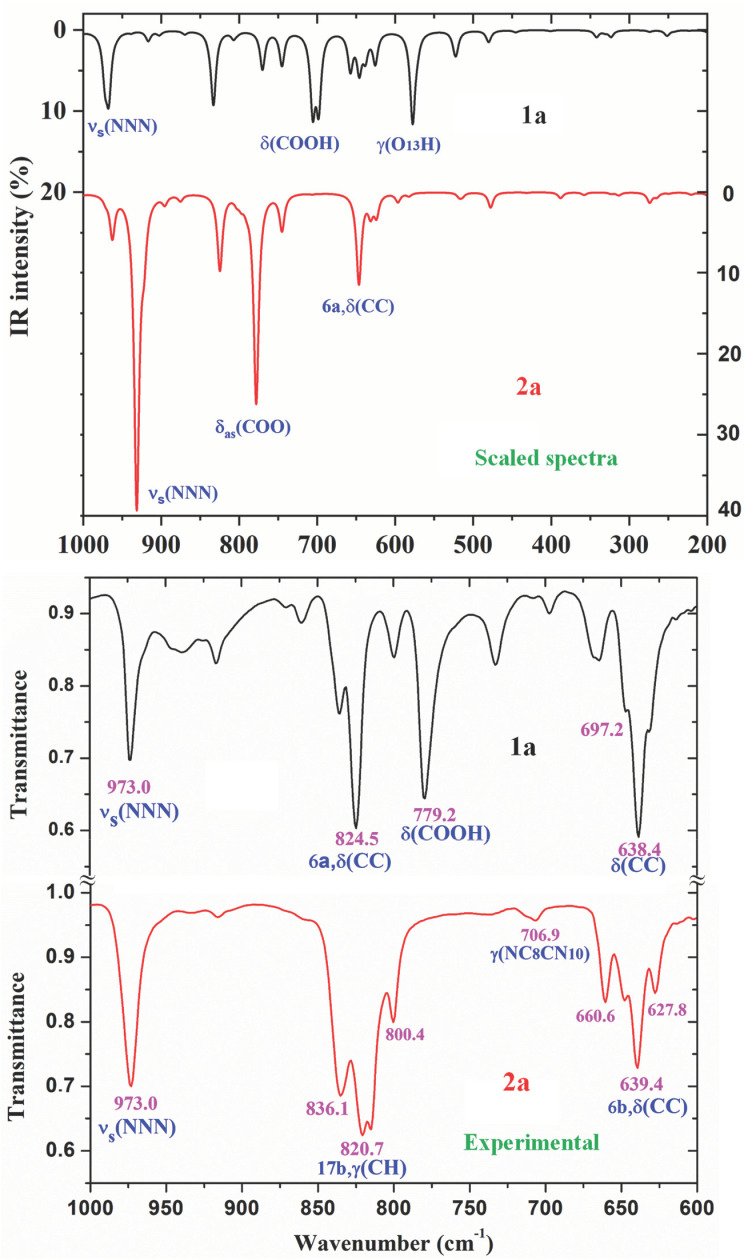
Comparison of the scaled IR spectra of **1a** and **2a** molecules by the PSE procedure in the 1000–200 cm^−1^ range with the experimental ones in the 1000–600 cm^−1^ range.

**Table 1 ijms-24-14001-t001:** Several selected optimized geometrical parameters calculated in the monomer form with the 6-31G(d,p) basis set. Bond lengths (r) are in Å, bond angles and dihedral angles (∠) are in degrees.

Parameters	0				−ΧΛ	−OH	0	0
	B3LYP		MP2		MP2	MP2	MP2	MP2
	1a	2a	1a	2a	1b	1c	1d	1e
r(C_4_−N_4_)	1.421	1.398	1.418	1.401	1.417	1.418	1.419	1.417
r(N_4_−N_7_)	1.349	1.357	1.340	1.348	1.340	1.340	1.34	1.34
r(N_4_−N_10_)	1.311	1.343	1.334	1.350	1.334	1.334	1.334	1.334
r(C_8_−C_9_)	1.440	1.448	1.422	1.427	1.423	1.422	1.422	1.424
r(C_9_−N_10_)	1.347	1.329	1.358	1.350	1.357	1.358	1.358	1.356
r(C_9_−C_11_)	1.465	1.549	1.466	1.541	1.467	1.466	1.466	1.468
O_12_…H_18_	2.152	1.980	2.181	2.030	2.186	2.181	2.18	2.188
∠(C_4_−N_4_−N_10_)	122.5	122.7	121.7	122.0	121.7	121.7	121.8	121.7
∠N−N−N)	115.9	115.0	116.7	116.1	116.8	116.7	116.7	116.8
∠(N_10_−C_9_−C_11_)	119.7	121.0	119.2	120.7	119.2	119.2	119.3	119.2
∠(C_9_−C_8_−N_14_)	133.2	131.0	132.6	131.0	132.7	132.6	132.6	132.7
∠(C_9_−C_11_=O_12_)	126.1	113.9	125.7	113.6	125.6	125.7	125.7	125.6
∠(C_9_−C_11_−O_13_)	112.4	115.7	111.9	115.3	111.8	111.9	111.9	111.8
∠(O=C=O)	121.4	130.3	122.4	131.0	122.6	122.4	122.4	122.6
∠(C_5_−C_4_−N_4_−N_10_)	−1.6	−1.1	−14.2	−3.0	−10.3	−13.7	−14.9	−10.7
∠(N_4_−N_10_−C_9_−C_11_)	175.6	175.3	173.6	176.1	173.7	173.6	173.6	173.7
∠(N_10_−N_4_−N_7_−C_8_)	−0.4	−1.1	−0.9	−1.4	−0.8	−0.9	−0.9	−0.8
∠(N_10_−C_9_−C=O_12_)	−165.9	−145.7	−157.1	−149.3	−156.3	−157.1	−157.3	−156.1
∠(N_10_−C_9_−C−O_13_)	13	32.8	21.9	29.8	22.6	21.9	21.7	22.9
∠(C_8_−C_9_−C=O_12_)	9.6	29.1	15.0	26.5	16.0	15.0	14.7	16.4
∠(C_11_−C_9_−C_8_−N_14_)	4.8	4.4	8.0	2.5	8.0	8.0	8.1	8
∠(C_9_−C_8_−N_14_−C_18_)	19.7	22.2	26.4	35.6	24.9	26.4	27	24.4
∠(C_8_−N_14_−C_15_−C_16_)	−163.0	−169.6	−151.8	−163.7	−152.0	−151.8	−151.7	−152.0
∠(N_14_−C_15_−C_16_−C_17_)	−21.5	−14.0	−22.9	−5.1	−23.6	−22.9	−22.6	−23.8

**Table 2 ijms-24-14001-t002:** Natural atomic charges (in *e*) calculated in the monomer form at the MP2/6-31G(d,p) level.

	−OCH_3_		−CL	−OH	−ONH_2_	−OCF_3_
Atom	1a	2a	1b	1c	1d	1e
O_1_	−0.606	−0.615	−0.012 *	−0.756	−0.472	−0.651
C_1_	0.378	0.332	−0.052	0.385	0.368	0.288
C_4_	0.117	0.167	0.160	0.115	0.117	0.163
N_4_	0.004	−0.064	−0.005	0.004	0.005	−0.007
N_7_	−0.380	−0.398	−0.381	−0.380	−0.380	−0.381
C_8_	0.458	0.409	0.462	0.458	0.457	0.462
C_9_	−0.099	0.019	−0.091	−0.099	−0.101	−0.089
N_10_	−0.188	−0.240	−0.183	−0.188	−0.185	−0.182
C_11_	0.979	0.956	0.979	0.979	0.979	0.978
=O_12_	−0.724	−0.889	−0.720	−0.724	−0.725	−0.719
O_13_	−0.776	−0.843	−0.775	−0.776	−0.776	−0.775
N_14_	−0.561	−0.558	−0.560	−0.561	−0.561	−0.560
C_18_	−0.211	−0.219	−0.211	−0.211	−0.211	−0.211
H_18_	0.263	0.297	0.263	0.263	0.263	0.263

* With CL.

**Table 3 ijms-24-14001-t003:** Molecular properties calculated at the B3LYP/6-31G(d,p) and M06-2X/6-31G(d,p) (values in parentheses) levels corresponding to **1a** and **2a** molecules.

Form	Molecular Properties		1a	2a
monomer	Rotational constants (GHz): A		0.616	0.647
		B	0.142	0.140
		C	0.116	0.116
	C_v_ (cal/mol·K)		70.04	69.04
	S (cal/mol·K)		144.58	144.01
	Dipole moment (Debye)		1.632	12.775
dimer	Rotational constants (GHz): A		0.114	(0.169)
		B	0.024	(0.034)
		C	0.021	(0.032)
	C_v_ (cal/mol·K)		143.7	(141.8)
	S (cal/mol·K)		252.9	(241.5)
	Dipole moment (Debye)		0.751	(9.734)

**Table 4 ijms-24-14001-t004:** Calculated harmonic wavenumbers (ν, cm^−1^), relative infrared intensities (A, %) and relative Raman scattering activities (S, %) obtained at the B3LYP/6-31G(d,p) level in **1a** and **2a**. Scaled (ν, cm^−1^) wavenumbers were obtained with the linear scaling equation procedure (LSE) and the polynomic scaling equation procedure (PSE). The main characterization of the different experimental bands was included. The number of the ring mode corresponds to Wilson’s notation [17].

ν^cal^ 1a	TLSE	PSE	A	S	Experimental 1a ^†^		Characterization of 1a *
	ν^scal^	ν^scal^			IR	Raman	
1670 1643 1605 1561 1515 1513 1476 1422 1407 1389 1298 1282 1269 1215 1207 1154 1075 995 990 825 672 652 640 591	1618 1592 1556 1514 1470 1468 1433 1382 1368 1351 1264 1249 1236 1185 1177 1127 1052 980 975 810 657 637 625 576	1625 1599 1562 1520 1476 1474 1438 1386 1372 1354 1266 1251 1238 1186 1178 1127 1050 973 968 807 657 638 626 578	9 2 100 40 22 17 5 2 12 2 64 3 13 3 5 14 12 5 7 1 4 3 4 11	100 6 17 48 6 10 1 57 46 0 0 2 2 0 1 0 0 9 11 0 0 1 1 1	1634.6 w 1594.0 vw 1565.1 vs 1514.0 vs 1482.2 vs 1463.9 s 1434.0 m 1381.9 m 1357.8 vw 1274.9 s 1265.2 s 1243.1 vs 1194.8 s 1153.4 vs 973.0 s 824.5vs 664.4 m 638.4 vs 614.3 vvw	1615.2 vs 1594.0 m 1561.2 m 1511.1 s 1485.1 w 1472.5 w 1420.5 m 1383.8 s 1362.6 vw 1357.8 vw 1262.0 m 1246.9 w 1243.0 w 1201.6 vw 1175.5 sh 1140.8 w 972.1 s 965.3 m 813.0 w 662.5 vw 639.2 m 631.7 m 570.9 w	8a, ν(C=C) (96) 8b, ν(C=C) (97) ν(C_8_-N_14_) (65) + ν_s_(N_7_CC) (20) 19a, ν(CC) (87) + δ_s_(CH) in pyrrolidine (11) δ_s_(C-H) out-of-phase pyrrolidine (83) δ_s_(C-H) out-of-phase pyrrolidine (75) 19b, ν(CC,CH) in aryl (72) ν_s_(NNN) (35) + ν_s_(C_4_N) (28) + ν(COO)(25) ν(C_4_N) (28) + ν(NNN) (22) + δ(COO) (18) ν(C_9_N) (32) +δ(COO) (25) + δ_s_(pyrrolidine) (23) ν(C-O1) (65) + 14, ν(CC) in aryl (22) γ_as_(C-H) in pyrrolidine (78) ν(NN)(53) + γ_as_(C-H) in pyrrolidine (33) γ_as_(C-H) in pyrrolidine (88) γ_as_(C-H) in pyrrolidine (82) + δ(O_13_H) (14) ν_s_(COOH) (45) + δ_as_(CH) in pyrrolidine (42) ν(O1-CH_3_) (83) _νas_(NNN) (42)+δ(CN_14_) (24) + γ(CC) pyrrolid (22) ν_s_(NNN) (38) + 12, δ(CC) (34) + δ(CC,CN) (24) 10a, γ(C-H) in aryl (97) γ_s_(triazole) (78) + γ(O13-H) (15) 6b, δ(CC) (42) + δ(triazole) (24) Γ(triazole) (34) + 6b, δ(CC) in aryl (33) γ(O_13_-H) (81)
**ν^cal^** **2a**	**TLSE**	**PSE**	**A**	**S**	**Experimental 2a** ^†^		**Characterization of 2a ***
	**ν^scal^**	**ν^scal^**			**IR**	**Raman**	
1674 1634 1591 1563 1514 1505 1435 1400 1391 1276 1213 1203 1085 994 953 722 638 610	1622 1583 1543 1516 1469 1461 1394 1361 1352 1243 1183 1174 1061 975 936 716 636 610	1629 1590 1549 1522 1475 1466 1399 1365 1356 1245 1184 1174 1060 972 932 706 624 597	3 1 58 100 1 15 0 19 8 79 8 8 10 1 32 0 2 1	100 2 12 23 2 13 20 7 0 1 0 1 4 2 6 0 0 0	1588.3 vs 1533.3 w 1514.0 vs 1474.5 m 1460.0 s 1392.5 w 1367.4 m 1346.2 m 1247.9 vs 1184.2 m 1173.6 m 973.0 m 933.4 vvw 706.9 vw 627.8 w 613.3 vvw	1610.4 vs 1592.1 m 1535.2 w 1515.9 vs 1476.4 m 1466.7 m 1393.4 vs 1372.2 vs 1356.8 w 1251.7 vw 1175.5 m 1063.7 w 975.0 vs 927.7 vw 707.9 w 627.8 w 607.6 w	8a, ν(C=C) (95) 8b, ν(C=C) (89) ν(C_8_-N_14_) (72) + ν_s_(N7CC) (15) 19a, ν(CC) (76) + δ_s_(CH) in pyrrolidine (18) δ_s_(C-H) out-of-phase in pyrrolidine (92) ν_s_(C_8_C_9_N) (62) + δ_s_(CH_2_) (16) + ν(NN) (15) ν(C_4_N) (41) + ν(NNN)(25) + 19a, ν(CC) (15) γ_s_(C-H) in-phase in pyrrolidine (87) ν_s_(NNN) (38) + γ_s_(C-H)pyrrolidine(30) + ν(C_8_C) (25) 7a,ν(COC) (62) + δ(CH) aryl (25) + γ_as_(CH_3_) (11) δ(triazole) (73) + γ(CC,CN) in pyrrolidine (16) γ_as_(C-H) in pyrrolidine (68) + δ(triazole) (15) ν_as_(COC) (62) + 15, δ(CH)(17) + γ_s_(CH_3_) (15) ν_as_(NNN) (32) + δ(CN_14_) (29)+δ(CC) pyrrolidi (25) ν_s_(NNN) (44) + ν(C_8_C) (20) + 10b, γ(CH) (18) γ(NC_8_CN10) (53) + γ_s_(COO) (38) Γ(triazole) (38) + δ(CC) in aryl (25) γ_s_(NNN) (55) + γ(COO) (18)

^†^ Observed frequencies characterized by notation: vs = very strong, s = strong, m = medium, w = weak band, vw = very weak, vvw = very very weak. * Notation used: ν_as_: anti-symmetric stretching, ν_s_: symmetric stretching, δ: in-plane bending, γ: out-of-plane bending.

## Data Availability

The data supporting information are available in the Appendix A.

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
