# Peer review of "Peculiarities of the Spatial and Electronic Structure of 2-Aryl-1,2,3-Triazol-5-Carboxylic Acids and Their Salts on the Basis of Spectral Studies and DFT Calculations"

_ijms, 2023, doi:10.3390/ijms241814001_

Round 1

Reviewer 1 Report

The Abstract is incomplete and must include information such as methodology, results and conclusions;

The conclusion is very long with long paragraphs. You must include the main results of this search.

In lines 32-33. Previously, the synthesis of a series of new 1,2,3-triazole derivatives with different substituents at N2-aryl ring, with the carboxylic (or carboxylate) group, in addition to the pyrrolidine group was reported by us [3]. Use impersonal terms in the writing of this manuscript, for example avoid the term "us" when citing your research group work;

In lines 34-41 it is necessary to reference the information;Nas linhas 43-57 é necessário referenciar as informações;

In line 67 . In the present paper, we have tested few substituents in para-position of the aryl ring, Figure 1. Avoid using personal terms like "we have tested";

The methodology is not very clear in the study;

It is an extensive study, but there are few bibliographic references in this research.

Author Response

I send it in the enclosed attachment

Reviewer 2 Report

Dear editor,

The paper contains a large set of conformation calculations on triazol derivatives. The lengthy enunciation of charge and geometric parameter relations results tedious and the explanations are not always tight, especially considering that there is no attempt to refer to any particular biological target.

On the other hand, there is a large section on attribution of spectroscopic features which may be interesting for people needing a characterization of their synthesis products, even partially in the solid state, where I deem the calculations insufficient.

Some places in the text require explanations or modification:

- 53: how would the formation of supramolecular structures (in solid?) affect pharmacological applications? please expand.

6-31G(d,p) may be enough for obtaining reliable structure parameters but not energies.

placement of double and aromatic bond in the pictures is very odd: not that there are really good ways to place them, but e.g. the difference between fig 2.a and fig 2.b is very strange.

130: "b3lyp fails in the calculation ...", now what does this mean? please detail how it fails here and again on line 132.

- 164: "the highest positive charge appears in the carbon atoms C8 and C11
because they are bonded to large negative atoms", what? on the contrary large charge separations need justification.

- 173-176: the dipole  moment is a vector, perhaps the authors should represent it along the molecule to make their discussion clearer.

- it is not clear to me if the authors expect the interaction with target to happen with the ligand in a neutral or ionic state.

- the authors do not discuss the solvent effects on the neutral and ionic forms of their molecules, nor which form might be more important in each biological environment.

- fig6 legend: " The two optimized dimer forms of 2a", I can see only one dimer form in the picture.

-234: "The anion 2a cannot be in cyclic dimer form in the crystal ...", what the authors are attempting to study here, is not clear to me. condensed phase must be neutral and therefore cannot be made of these anions alone, at least a counterion must be conceived. I do not understand what they mean by staking interactions, perhaps stacking? Stacking is necessary even for the neutral molecule: how do the authors think the neutral dimers lie one above the other? Unfortunately, this is not the proper way to study the solid system, they need a software with translational symmetry capabilities. Using such software, they would discover that the structure is most probably quite different from their expectations and, more importantly here, from their results.

- 281: " In the dimer forms, the dipole moment value is slightly lower in 1a than in its monomer", no, it is much lower, also it points in an entirely different direction. what is this idea of showing the b3lyp values where available and the m06-2x where the b3lyp are not available? this cannot be allowed. And again, b3lyp is only a function, it cannot fail, it is the software that is failing.

the paper is well structured but the language is somewhat loose. I signal below the points where poor language may impair a correct understanding; I also recommend a thorough reading on behalf of the authors.

- 34: "two of which" is too far from derivatives, to which it refers

- 49: "optimized" is not clear in the present context

- 53: "affect to" -> "affect"

- 61: "an improvement was raised" ???

- 69: adjust the verbs position and tense

- 78: "their values", which values of what?

- 148: "the loose" ???

- 148-150: it seems to me that indeed the same concept is repeated twice. indeed it seems to me that the concept was repeated twice.

Author Response

I send it in the enclosed attachment

Reviewer 3 Report

The manuscript by Alcolea Palafox et al reports the Peculiarities of the spatial and electronic structure of 2-aryl-1,2,3-triazol-5-carboxylic acids and their salts on the basis of spectral studies and DFT calculations. Triazole-containing molecules are important pharmacophores in medicinal chemistry especially in drug discovery. The authors explored the electronic effect of five substituents in para-position of the aryl ring on the molecular structure of the triazole and atomic charge distribution was determined. Moreover, Anticancer activity was studied by 1,2,3-triazoles derivatives from the structural and spectroscopy points of view. The manuscript was well organized and well written. I recommend publication in the IJMS.

Author Response

Thanks a lot for the acceptation

Reviewer 4 Report

In the paper, the vibrational analyses of the FT-IR and FT-Raman of 2-aryl-1,2,3-triazol-5-carboxylic acid and its salt have been reported. The paper is quite accurate and correct. Anyway, the manuscript is not a highly original work, surely not for the high quality of IJMS. In fact, the authors prepared and studied only one compound and its salt for recording only FT-IR and FT-Raman spectra and then assigning almost all the peaks with the aid of software. No applications of the compounds are reported. For this reason, for me in this form, the manuscript does not reach the high target of IJMS.

Further comments

1.       The keyword section must not include “anti-cancer drugs”.

2.       For Figure 2b, the bonds in the COO group must be delocalized bonds.

3.       There are no benchmark values for bond lengths, bond angles, and torsion angles reported in Table 1. It is recommended to report the values from the experimental data (e.g., SC-XRD) for better comparison.

4.       For the “Chloro” substituent, use Cl, not CL.

5.       Please check whether -ONH2 functional group is correct or not, as this functional group is not very common in synthetic organic chemistry.

6.       In my opinion, the results regarding the electronic structures are too few in this recent manuscript. However, the title of the manuscript includes the word “electronic structure”.

7.       There is an inconsistency in the decimal places reported throughout the manuscript. However, no decimal place must be reported as the most common resolutions of the FT-IR and FT-Raman is 1-4 cm-1.

8.       For each vibrational mode, it is not necessary to mention the atomic labeling number. For instance, C-O modes not C-O13 modes.

9.       No software reference for the vibrational analysis and %PED calculation (e.g. VEDA software) is cited.

10.   NMR, MS, and elemental analysis results must be included in the supporting information section.

 Minor editing of English language is required

Author Response

I send it in the enclosed attachment

Round 2

Reviewer 1 Report

Improvements were made to the manuscript. I agree with the publication.

Reviewer 4 Report

The revised paper can be accepted for publication.